# High-Performance Pockels Effect Modulation and Switching in Silicon-Based GaP/Si, AlP/Si, ZnS/Si, AlN/3C-SiC, GaAs/Ge, ZnSe/GaAs, and ZnSe/Ge Superlattice-On-Insulator Integrated Circuits

**DOI:** 10.3390/s22207866

**Published:** 2022-10-16

**Authors:** Francesco De Leonardis, Richard Soref

**Affiliations:** 1Photonics Research Group, Department of Electrical and Information Engineering, Politecnico di Bari, Via Orabona 4, 70126 Bari, Italy; 2Department of Engineering, University of Massachusetts at Boston, Boston, MA 02125, USA

**Keywords:** photonic integrated circuit, micro-ring resonator, Pockels effect, superlattice, electro-optic modulator, optical switch, IoT, sensor networks

## Abstract

We propose new a Si-based waveguided Superlattice-on-Insulator (SLOI) platforms for high-performance electro-optical (EO) 2 × 2 and N × M switching and 1 × 1 modulation, including broad spectrum and resonant. We present a theoretical investigation based on the tight-binding Hamiltonian of the Pockels EO effect in the lattice-matched undoped ${\left(\mathrm{GaP}\right)}_{N}/{\left({\mathrm{Si}}_{2}\right)}_{M}$, ${\left(\mathrm{AlP}\right)}_{N}/{\left({\mathrm{Si}}_{2}\right)}_{M}$, ${\left(\mathrm{ZnS}\right)}_{N}/{\left({\mathrm{Si}}_{2}\right)}_{M}$, ${\left(\mathrm{AlN}\right)}_{N}/{\left(3C-\mathrm{SiC}\right)}_{M}$, ${\left(\mathrm{GaAs}\right)}_{N}/{\left({\mathrm{Ge}}_{2}\right)}_{M}$, ${\left(\mathrm{ZnSe}\right)}_{N}/{\left(\mathrm{GaAs}\right)}_{M}$, and ${\left(\mathrm{ZnSe}\right)}_{N}/{\left({\mathrm{Ge}}_{2}\right)}_{M}$ wafer-scale short-period superlattices that are etched into waveguided networks of small-footprint Mach-Zehnder interferometers and micro-ring resonators to yield opto-electronic chips. The spectra of the Pockels **r_33_** coefficient have been simulated as a function of the number of the atomic monolayers for “non-relaxed” heterointerfaces. The large obtained **r_33_** values enable the SLOI circuit platforms to offer a very favorable combination of monolithic construction, cost-effective manufacturability, high modulation/switching speed, high information bandwidth, tiny footprint, low energy per bit, low switching voltage, near-IR-and-telecom wavelength coverage, and push-pull operation. By optimizing waveguide, clad, and electrode dimensions, we obtained very desirable values of the *V_π_L* performance metric, in the range of 0.062 to 0.275 V·cm, portending a bright future for a variety of applications, such as sensor networks or Internet of Things (IoT).

## 1. Introduction

This paper presents a detailed analysis of waveguided electro-optical (EO) modulators and switches within seven new superlattice-on-insulator (SLOI) structures. The large-area SLOI wafer is a recently proposed [1,2] CMOS-compatible optoelectronics (OE) platform that is projected to enable many key applications in computing, communications, sensing, and quantum photonics. The initially uniform short-period superlattice stack is etched into a wafer-scale array of photonic integrated circuits (PICs) containing efficient Pockels networks. Ideally, the control electronics are integrated onto the same wafer and, thus, this OE platform becomes the source of versatile, high-performance chips.

The SLOI wafer is believed to be compatible with high-volume foundry manufacture. We have proposed [2] to create the large-diameter foundry wafer by direct wafer bonding of a superlattice donor wafer to an oxidized silicon receiver wafer having, for example, 300 mm diameter. The donor is cut back after bonding, leaving only the uniform SL stack. This (${\left(A\right)}_{N}/{\left(B\right)}_{M}$) stack consists of *N* monolayers (MLs) of material *A* (group III–V) and *M* monolayers of material *B* (group IV or III–V).

This paper focusses on the engineering and optimization of the linear EO effect (the Pockels effect) in seven undoped, lattice-matched short-period superlattices (SPSLs) that provide fast, low-energy, low-loss, and waveguided EO modulation (broad spectrum or resonant modulators) as well as broad spectrum switching “meshes” (such as neuromorphic meshes, beam-steering meshes, and N × M spatial routing matrices) operating at visible-light wavelengths in addition to telecom wavelengths. The results found here indicate that the SLOI chips will be fully competitive with Pockels chips formed from lithium-niobate-on-insulator, barium-titanate-on-insulator, and organic-polymer hybridized with silicon-on-insulator.

Our recent investigations [3] showed that very strong second-order and third-order nonlinear optical responses can be engineered in the GaP/Si SLOI, and second-order nonlinearity is the basis of the very large Pockels effects that we predict. By choosing the atomic layering of the SPSL properly, one wafer can feature excellent nonlinear and linear functions; that is, the SLOI PIC can provide quantum light sources (or classical harmonics) as well as leading-edge Pockels devices.

If we consider for a moment the SPSLs AlN/GaN, AlAs/GaAs, and GaSb/InAs, it is true that the scientific literature provides experimental results on these hetero-materials, and it is encouraging that smooth nano-layering was achieved [4,5,6]. On the other hand, the particular SPSL materials area spotlighted here are a speculative area of research. Theory awaits experiment. Molecular-beam epitaxy (MBE) experiments have not yet been performed to prove that these seven superlattices can be fabricated with high crystalline qualities for the 300 to 500 layers of the stack. Therefore, the work here provides only theoretical guidelines on what SLOI performance can be expected if the layering is “nearly perfect.” Of course, in developmental experiments there will be fabrication errors including surface roughness, etching errors, variation in layer thicknesses, point defects, threading dislocations, and perhaps unwanted strain as well as unintentional doping.

Such errors will induce propagation loss within the passive SLOI waveguides and will decrease the performance metrics for the SLOI EO and nonlinear optical (NLO) devices. We have not performed a quantitative theoretical analysis of the detriments caused by such errors. Instead, we are just indicating what will be achievable if and when the fabrication problems are solved. We are not saying that those problems will be solved. Rather, we are, in some sense, offering motivation for the fabrication challenges to be addressed. By avoiding doping the layers and not fabricating strained layers, our SL choices here simplify the MBE task.

The paper is organized as follows. Motivations for realizing Photonic Integrated Circuits (PICs) based on SLOI are detailed in Section 2. The theory background is reported in Section 2. Numerical results about the r_33_ calculations are detailed in Section 3. The large coefficients that are found are deployed to demonstrate the potential of the proposed platform for inducing state-of-the-art EO modulators (both resonant and broad modulators). Moreover, the uniquely valuable SLOI potential for 2 × 2 crossbar switching are quantified. Combined EO nonlinear and PIC operation is discussed in Section 4, while Section 5 specifies a new method for integrating photodetectors on the SL waveguide. Finally, Section 6 summarizes the work.

## 2. Materials and Methods

In this section we present in detail the short period superlattice structures and the mathematical model adopted in order to calculate the Pockels coefficient.

### 2.1. Superlattice-on-Insulator Photonics

We focus on the following undoped SPSLs: ${\left(\mathrm{GaP}\right)}_{N}/{\left({\mathrm{Si}}_{2}\right)}_{M}$, ${\left(\mathrm{AlP}\right)}_{N}/{\left({\mathrm{Si}}_{2}\right)}_{M}$, ${\left(\mathrm{ZnS}\right)}_{N}/{\left({\mathrm{Si}}_{2}\right)}_{M}$, ${\left(\mathrm{AlN}\right)}_{N}/{\left(3C-\mathrm{SiC}\right)}_{M}$, ${\left(\mathrm{GaAs}\right)}_{N}/{\left({\mathrm{Ge}}_{2}\right)}_{M}$, ${\left(\mathrm{ZnSe}\right)}_{N}/{\left(\mathrm{GaAs}\right)}_{M}$, and ${\left(\mathrm{ZnSe}\right)}_{N}/{\left({\mathrm{Ge}}_{2}\right)}_{M}$, in which the *N* monolayers of the group III–V material and the *M* monolayers of the group IV or III–V material are repeated periodically along the [1 1 1] growth direction (*z* axis). Note that the single ML is comprised of a two-atom-thick layer (i.e., cation Ga and anion P). Moreover, all atoms are on the sites of a zinc-blende lattice with lattice constant *a_L_* (under the perfect-matched condition). The *x* and the *y* coordinate axes are chosen along [$\bar{2}$ 1 1] and [0 $\bar{1}$ 1]. In order to realize the waveguiding structures such as straight waveguides, Mach-Zehnder interferometers (MZIs), and micro-ring resonators (MRRs) where optical mode confinement within the SPSL is needed. Thus, we propose to first grow the SPSL stack using molecular-beam epitaxy (MBE) [7] upon a large-area lattice-matched donor wafer such as 300 mm diameter silicon or Ge-buffered silicon. The number of periods of layering, depending on the particular values of *a_L_*, *N*, and *M*, is chosen in order to provide a uniform stack thickness across the wafer, a thickness able to support fundamental optical modes at the operation wavelength (in the visible or near IR or mid IR). After that, the donor is bonded to an oxidized silicon receiver wafer having about 2 μm of top SiO_2_. Next, the material above the SL is cut away to leave only the stack, thus creating a superlattice-on-insulator (SLOI) structure. Finally, an etching process is applied in order to create the fundamental photonic devices based on low-loss SPSL strip and rib waveguides, and the connected devices are configured into PICs with complex functionalities.

We recently quantified that the GaP/Si SL system can enable the possibility of realizing giant $\chi_{zzz}^{\left(2\right)}$ susceptibility. We proposed in [8] a multi-period stack of two doped asymmetric coupled quantum wells (ACQWs), where giant values of $\chi_{zzz}^{\left(2\right)}$ are induced by the combination of the dopant surface density and by the double resonance condition obtained via engineering the first three quantum-confined states. There, we explored the feasibility of realizing giant $\chi_{zzz}^{\left(2\right)}$ susceptibility in n-doped [3] and undoped [2] ${\left(\mathrm{GaP}\right)}_{N}/{\left({\mathrm{Si}}_{2}\right)}_{M}$ SPSLs, where the coefficient $\chi_{ijk}^{\left(2\right)}\left(2\omega ,\omega ,\omega \right)$ was calculated considering the electron transitions inside the conduction band (CB) and between the valence band (VB) and CB, respectively.

Because of the large susceptibilities *χ*^(2)^ and *χ*^(3)^ found in our initial SPSL, we expect to find large Pockels effects generally in SPSL structures, making the waveguided SLOI platform a good candidate for both classical and quantum applications. Indeed, efficient nonlinear effects such as Second Harmonic Generation, Spontaneous Parametric Down Conversion, and Spontaneous Four Wave Mixing and efficient electro-optic modulation can be realized on the same technological platform. In the paragraphs below, we demonstrate that the Pockels coefficients of the SPSLs proposed here are lower than those of BaTiO_3_ [9], but much larger that the values recorded for LiNbO_3_ and non-centrosymmetric semiconductors.

Moreover, within the monolithic semiconductor waveguides, we think the metrics of our EO modulators can be very competitive with the metrics obtained in silicon PN carrier-depletion modulators.

For both classical and quantum applications, future advances in PICs will likely require a heterogeneous approach [10] that combines multiple materials in order to optimize light generation, optical amplification, modulation, switching, routing, qubit input/output (I/O) interfacing, passive splitting/combining, filtering, frequency translation, and photo-detection. In principle, our SLOI platforms could contribute to all of these areas (except perhaps lasing and gain).

### 2.2. Theoretical Background

In this section we describe the mathematical modeling for the calculation of the Pockels coefficient, generated in short-period superlattices based on ${\left(\mathrm{GaP}\right)}_{N}/{\left({\mathrm{Si}}_{2}\right)}_{M}$, ${\left(\mathrm{AlP}\right)}_{N}/{\left({\mathrm{Si}}_{2}\right)}_{M}$, ${\left(\mathrm{ZnS}\right)}_{N}/{\left({\mathrm{Si}}_{2}\right)}_{M}$, ${\left(\mathrm{AlN}\right)}_{N}/{\left(3C-\mathrm{SiC}\right)}_{M}$, ${\left(\mathrm{GaAs}\right)}_{N}/{\left({\mathrm{Ge}}_{2}\right)}_{M}$, ${\left(\mathrm{ZnSe}\right)}_{N}/{\left(\mathrm{GaAs}\right)}_{M}$, and ${\left(\mathrm{ZnSe}\right)}_{N}/{\left({\mathrm{Ge}}_{2}\right)}_{M}$, technological “on insulator” platforms.

Microscopically, the Pockels coefficient is a tensor, *r_ijk_*, that relates the change in the component *ε_ij_* of the inverse optical dielectric tensor of a non-centrosymmetric crystal to a static (or low frequency), external electric field applied in the *k*-direction (*E_k_*), according to the following equation [11,12]:(1)$$\Delta {\left(\varepsilon^{-1}\right)}_{ij}=\sum_{k}r_{ijk}E_{k}$$

By neglecting any modification of the unit cell shape due to the inverse piezoelectric effect, the EO tensor is only the sum of an electronic ($r_{ijk}^{\left(el\right)}$) contribution and an ionic ($r_{ijk}^{\left(ion\right)}$) contribution. The electronic contribution is due to an interaction of the electric field with the valence electrons when considering the ions artificially clamped at their equilibrium positions. It is proportional to the second-order nonlinear optical susceptibility $\chi_{ijk}^{\left(2\right)}\left(\omega ,\omega ,0\right)$, according to Equation (2) [12,13,14]:(2)$$r_{ijk}^{\left(el\right)}=-\frac{8\chi_{ijk}^{\left(2\right)}\left(\omega ,\omega ,0\right)}{n_{i}^{2}n_{j}^{2}}$$
where *n_i_* and *n_j_* are the principal refractive indices of the SPSL.

The calculations of *n_i_*, *n_j_* and $\chi_{ijk}^{\left(2\right)}\left(\omega ,\omega ,0\right)$ are performed within the tight-binding (TB) framework. It describes the SPSL system by means of a real-space Hamiltonian-matrix function (*H*(**k**)), of a reduced number of parameters, resulting in a reduction of the computational costs from the ab initio methods. The motivations for this choice and the calculation details are well presented in our previous works [3], where the *H*(**k**) Hamiltonian of two different alternating zinc-blende crystals, labelled *ca* and *CA* (where c (C) and a (A) are the cation and the anion, respectively) is represented as a block matrix. The number of block matrices is strictly related to the MLs numbers *N* and *M* The block matrices are 10 × 10 matrices denotated as *H_ca_*_(*CA*)_, *G_ca_*_(*CA*)_, and *F_ca_*_(*CA*)_ and representing the intramaterial interaction for (*ca*)*_N_* and (*CA*)*_M_*, respectively. Moreover, the matrices *G_i_* and *F_i_* (*i* = 0, 1) are included in order to describe the intermaterial interactions. In this context, we first calculate the electronic structure of the SPSL by diagonalizing the Hamiltonian *H*(**k**) and consequently we calculate the dipole matrix elements needed to evaluate the terms *n_i_*, *n_j_*, and $\chi_{ijk}^{\left(2\right)}\left(\omega ,\omega ,0\right)$ [2,11].

By definition, the dipole matrix element is provided by: 〈n,k|u·r|m,k〉 = 〈u·n,k|r|m,k〉, where ***u*** is the polarization vector of the electric field and **r** is the vector position operator. Here, *n* and *m* denote the band index, running in either VB or CB, and ***k*** indicates the wavevector. As detailed in [15,16,17,18], in *k*-space the position operator is proportional to the gradient with respect to k of the Hamiltonian *H*(**k**):(3)$$\mu_{nm}=\langle n,k|r|m,k\rangle =\frac{j}{\left(E_{m}\left(k\right)-E_{n}\left(k\right)\right)}\langle n,k|\nabla_{k}H\left(k\right)|m,k\rangle$$
where *E_i_*(**k**) represents the energy at the **k** point for the *i*-th band.

The ionic contribution to the EO response depends upon the relaxation of the atomic positions due to the applied quasistatic electric field, which, in turn, induces the variations on the tensor elements *ε_ij_*. Rigorously speaking, the term $r_{ijk}^{\left(ion\right)}$ is related to the Raman susceptibility tensor, as described in [13]. However, it requires infrared Raman spectroscopy in order to evaluate the phonon mode dispersion curves. Due to the novelty of the superlattice platforms adopted here, no experimental data are present in the literature. Thus, to meet the purpose of predicting the properties of new nanocomposite materials such as the SPSLs proposed here, the theory is required to employ as few physical parameters as possible. In this context, we adopt the semiclassical approach based on the energy band diagram, the dielectric theory, and the concepts of bond charge and effective ionic charge. As indicated in [12], the band gap *E_g_* (evaluated here by means of the TB method [3]) can be decomposed into the covalent (*E_h_*) and ionic ($E_{C}$) parts. The contribution *E_h_* depends only on the internuclear spacing, while the ionic gap $E_{C}$ is provided using a screened Coulomb interaction. On the basis of the estimations of both *E_h_* and $E_{C}$, we calculate the contribution $r_{ijk}^{\left(ion\right)}$ by means of Equation (14.44) of [12], which accounts reasonably well for both the magnitude and sign of $r_{ijk}^{\left(ion\right)}$ in zinc-blende and wurtzite crystals. We are aware that the above theory is based on a variety of simplifying assumptions. However, since the foundations of this theory are empirical, some of the physical parameters involved in the model can be better set on the basis of future experimental measurements.

## 3. Numerical Results

In this section, we present the numerical results on the Pockels coefficient obtained in superlattice material systems. Moreover, the performance metrics of both MZI and MRR electro-optic modulators based on SLOI waveguides will be derived as a function of the calculated r_33_ coefficient. In this context, the cross-section of one waveguide and the geometrical parameters are sketched in Figure 1a, where the SPSL ${\left(A\right)}_{N}/{\left(B\right)}_{M}$ represents any SL platform among ${\left(\mathrm{GaP}\right)}_{N}/{\left({\mathrm{Si}}_{2}\right)}_{M}$, ${\left(\mathrm{AlP}\right)}_{N}/{\left({\mathrm{Si}}_{2}\right)}_{M}$, ${\left(\mathrm{ZnS}\right)}_{N}/{\left({\mathrm{Si}}_{2}\right)}_{M}$, ${\left(\mathrm{AlN}\right)}_{N}/{\left(3C-\mathrm{SiC}\right)}_{M}$, ${\left(\mathrm{GaAs}\right)}_{N}/{\left({\mathrm{Ge}}_{2}\right)}_{M}$, ${\left(\mathrm{ZnSe}\right)}_{N}/{\left(\mathrm{GaAs}\right)}_{M}$, and ${\left(\mathrm{ZnSe}\right)}_{N}/{\left({\mathrm{Ge}}_{2}\right)}_{M}$. In addition, Figure 1b and Figure 2b show a broad-spectrum 1 × 1 EO modulators based on push-pull MZI and ring resonator architectures, respectively. A localized rib-waveguide structure in the push-pull region is assumed. The other waveguides are strips. Because of the planarized SiO_2_ upper cladding that covers all waveguides, the electrode configuration is assumed to be coplanar, where the signal electrode is placed on the top of the rib waveguide and the two ground electrodes are on the sides of each waveguide. Moreover, Figure 3 shows the top view of the broad-spectrum 2 × 2 MZI crossbar switch in the SLOI platform.

### 3.1. Pockels Coefficient in Superlattice Platform

We perform the $r_{ijk}=r_{ijk}^{\left(el\right)}+r_{ijk}^{\left(ion\right)}$ calculations for all combinations of *N* and *M* satisfying the condition *N* + *M* = 3 *N* + *M* = 6 and *N* + *M* = 9. The *C*_3*v*_ point group is expected for a SPSL grown in the [1 1 1] direction [3].

The dominant electro-optic tensor element is r_33_ (Voigt notation) and for this reason it will be the object of the following analysis. Hereafter, we will consider the ${\left(\mathrm{GaP}\right)}_{N}/{\left({\mathrm{Si}}_{2}\right)}_{M}$ system as our reference SPSL platform, since very large second- and third-order nonlinear effects (i.e., $\chi_{zzz}^{\left(2\right)}\left(2\omega ,\omega ,\omega \right)$ and $\chi_{xxxx}^{\left(3\right)}\left(3\omega ,\omega ,\omega ,\omega \right)$) have been demonstrated in our recent works [2,3]. In this context, Figure 4 shows the r_33_ coefficient as a wavelength function for several values of *N* and *M*. In each case the simulations have been performed for wavelengths larger than the inter-band edge wavelength (such as 822 nm), where the absorption losses are negligible.

We record a large value of the r_33_ coefficient of 89.3 pm/V at 822.5 nm for the SPSL ${\left(\mathrm{GaP}\right)}_{2}/{\left({\mathrm{Si}}_{2}\right)}_{1}$. We have performed similar calculations for all of the above-mentioned superlattice platforms. The results of our investigations are summarized in Figure 5, where for each SL we have reported only the cases providing the larger r_33_ at the band-edge wavelength. It is interesting to note that very large values of r_33_ = 103.5 pm/V and 114.7 pm/V have been obtained for ${\left(\mathrm{ZnS}\right)}_{7}/{\left({\mathrm{Si}}_{2}\right)}_{2}$ and ${\left(\mathrm{ZnSe}\right)}_{2}/{\left(\mathrm{GaAs}\right)}_{7}$, respectively. Moreover, all the curves indicate that values of r_33_ larger than those of the bulk semiconductors can be obtained, enabling the SLOI platform to be a good candidate for realizing high-performance EO modulators for wavelengths ranging from the 500 nm visible to the near-infrared and through the mid-infrared region.

Figure 6 shows the larger values of the r_33_ coefficient, calculated at 1550 nm and for all the SL platforms considered.

It is worth outlining that in all r_33_ calculations, both the Hamiltonian *H*(**k**) and the dipole matrix elements are calculated taking into account the valence band offset parameter, Δ*E_v_*. The Δ*E_v_* values used in the simulations are listed in Table 1 for all the considered SL platforms.

The results provided here have been obtained assuming that the two crystals are lattice-matched with unrelaxed interfaces. However, if heterointerface bonds with length *d* are stretched (strained) by the amount Δ, then the inter-material interaction energies decrease according to Harrison’s scaling rule. As a result, the elements of the block matrices *G_ca_*_(*CA*)_ and *F_ca_*_(*CA*)_ [2,3] are reduced by a factor (1 + Δ*d*)^−2^. We have analyzed the effects of both the bond relaxation of the anion of (*ca*)*_N_* to the cation (*CA*)*_M_* and of the anion of (*CA*)*_M_* to the cation (*ca*)*_N_* and we have recorded that only one of these induces detrimental effects on the Pockels coefficient. To provide a specific example, by considering ${\left(\mathrm{GaP}\right)}_{2}/{\left({\mathrm{Si}}_{2}\right)}_{1}$, and assuming Δ/*d* = 0.1, we observe a r_33_ reduction of 39% for heterointerface bond relaxation of P-on-Si, and 3.2% reduction for Si-on-Ga bond relaxation.

### 3.2. EO 1 × 1 Modulators Based on the SLOI Platform

Due to the electro-optical modification of the refractive index in a Pockels material, light undergoes a controllable modulation in its optical phase as it travels through the medium. This is electro-refraction (phase shifting) without electro-absorption. This shifting is utilized in electro-optic devices, where the device architecture is set on the basis of the specific application. Commonly, broad-spectrum Mach-Zehnder interferometers in the 1 × 1 configuration are used in order to induce amplitude modulation. In particular, MZIs allow push-pull operation, where the voltage is applied across both arms with an equal magnitude but with an opposite sign with respect to each other. Then, the interferometer arms experience opposite phase shifts, inducing a *π*-phase difference over a modulation length (*L*) scaled by a factor of 2 when compared with the single-arm operation at the same driving voltage. Generally, the overall performance of EO devices not only depends on the Pockels coefficients but also on the confinement of light to the active electro-optic material and the electro-optic field overlap. In this sense, Figure 7 shows both the spatial distribution of |***E***|^2^ for the fundamental TM mode and the applied static electric field *E_RF_* (black arrows). Our simulations are based on full-vectorial FEM calculations where a multiphysics approach has been adopted. The FEM electromagnetic module that is used to evaluate the optical mode distributions inside the active region of the EO modulator, works together with the FEM static module in order to simulate the distribution of the applied static electric field and the capacitance per unit length (*C*/*L*) of the EO structure. Under this scenario, the overlap between the optical and static electric fields is calculated in order to better evaluate the product *V_π_L* (see Equations (4) and (5)).

Ideally, in Figure 7, the RF field should be vertical in the core of the SL waveguide to maximize the r_33_ response, but the field direction in practice deviates from the ideal orientation.

The half-wave voltage-length product *V_π_L* has emerged as a useful figure of merit (FOM). An ultimate goal of a modulator design is to achieve the smallest possible *V_π_L*, resulting in the smallest footprint with the least required voltage. Moreover, a reduced *V_π_L* product leads to further performance benefits with respect to modulation bandwidth and energy consumption. A shorter interaction length induces a smaller device capacitance *C* (due to a decreased electrode area), which in turn determines larger bandwidth *f*_3*dB*_ and lower electric power consumption *E_bit_* according to: $f_{3dB}=1/2\pi RC$ and $E_{bit}=CV^{2}/4$, respectively.

Assuming a symmetrical push-pull MZI (see Figure 1b), the *V_π_L* is provided by [19]:(4)$$V_{\pi }L=\frac{\lambda n_{eff}}{2\pi n_{z}^{4}\Gamma }$$
where *n_eff_* is the effective index of the waveguide mode (TM in our case) and *n_z_* the material index of SL structure (see Figure 1a). The term Γ is an overlap integral parameter provided by [19]:(5)$$\Gamma =\frac{\iint \left[E_{RF}\left(x,y\right)/V\right]r_{33}\left(x,y\right){\left|E\left(x,y\right)\right|}^{2}dxdy}{\iint {\left|E\left(x,y\right)\right|}^{2}dxdy}$$
where *E_RF_*/*V* is the RF frequency electric field per volt applied to the electrodes, and *E* is the electric field of the fundamental TM optical mode.

A further figure of merit is the optical insertion loss IL, induced by the loss coefficient *α* inside the integrated waveguide. Our expectation is that the IL of any SLOI passive (non-electroded) waveguide, for all seven cases, will be sufficiently below 1 dB/cm, although unwanted surface-scattering loss could affect the “overall IL” if the fabrication process is not well controlled.

For EO modulators and switches, we need to minimize the electrode-induced losses (*α_el_*), in order to keep the total propagation loss below 1 dB/cm. Our preliminary simulations indicate that *α_el_* is weakly dependent on the electrode gap (*G*), but it is strongly influenced by the geometrical parameters *d* (see Figure 1a). In this context, Figure 8 plots the contribution *α_el_* as a function of the parameter *d*, for the SPSL ${\left(\mathrm{GaP}\right)}_{2}/{\left({\mathrm{Si}}_{2}\right)}_{1}$, and assuming: *W* = 700 nm, *H* = 300 nm (corresponding to 320 SL periods), *h* = 150 nm (corresponding to 159 SL periods), *G* = 2000 nm, and *λ* = 822.5 nm.

The plot clearly indicates that values of *d* larger than 500 nm are required in order to keep *α_el_* lower than 0.1 dB/cm (considered negligible with respect to the contributions induced by both the fabrication process and surface roughness).

Under this scenario, the figures of merit of the MZI push-pull modulator, sketched in Figure 1b, are shown in Figure 9, where the product *V_π_L* and *C*/*L* are plotted as a function of the electrode gap, *G*. The minimum in the *V_π_L* plot is induced by the maximum in the integral overlap Γ.

The plot records a minimum of *V_π_L* = 0.096 Vcm at *G* = 1000 nm. Moreover, the parameter *C*/*L* decreases, increasing the electrode gap. For example, setting *G* = 2000 nm, as a trade-off choice, we record the following performances for a modulation length *L* = 2 mm [20]: *V_π_* = 0.53 V, *C* = 60.52 fF, *f*_3*dB*_ = 52.6 GHz, and *E_bit_* = 4.21 fJ/bit. With short modulation length, the electrodes can be treated as lumped elements driven as capacitors. The modulation bandwidth is, thus, limited by the product of the capacitance (*C*) and total resistance (*R*), currently limited by the impedance of the network analyzer drive (50 Ω). However, we can obtain modulation bandwidth at a much higher value (>100 GHz) using a traveling wave design, where the bandwidth is influenced by the velocity mismatch between optical and RF signals and by metal RF loss [19]. However, this approach is not considered here, since we are interested in a modulator design minimizing the footprint. All of the simulations in this paper have been performed considering gold electrodes. However, we have also investigated how the FOM of each device is affected by deploying CMOS-compatible electrode metals upon the waveguides. This is illuminated by the following examples: fixing *d* = 385 nm, *G* = 850 nm, *λ* = 822.5 nm, and taking the ${\left(\mathrm{GaP}\right)}_{2}/{\left({\mathrm{Si}}_{2}\right)}_{1}$ SPSL, we record *V_π_L* of 0.080 V cm, 0.083 V cm, and 0.155 V cm for Au, Cu, and Al electrode metals, respectively. Thus, we conclude that Cu will yield performances moderately similar to those presented here.

Using Figure 9 procedure, and taking *L* = 1 mm, we performed a two-parameter minimization of the *V_π_L* for all seven platforms, determining for each the combination of *d* and *G* values that gave the optimum FOM, with the *d* value being chosen under the assumption of 1 dB/cm loss for the active region of the modulator or switch; the results are displayed in Table 1. The 0.062-0.275 FOMs are competitive with those of LiNbO_3_ and BaTiO_3_. (Table 2 of [21]). Here, the switching voltage *V_s_* is the same as the *V_π_* voltage. The *V_s_* is found to range from 0.68 V to 2.75 V at the wavelength shown, and this will yield a high extinction ratio, such as 15 dB in the modulators. If we could accept 7 dB of extinction, for example, then the modulation voltage would be about one-half of what is listed. Note that the IL is quite small in each case. Bandwidths exceed 72 GHz and energy per bit is in the fJ/bit range.

At this step, we will analyze the performances of the EO modulators based on MRR architecture (see Figure 2b) taking again the 1 dB/cm assumption. In this context, we need to apply an external voltage (*V*) that is large enough to cause a resonance shift of Δλ=1×δλ or Δλ=2×δλ, where *δλ* represents the 3 dB resonance linewidth. Moreover, the total switching delay is defined as $\tau_{switch}\;$= $\tau_{RC}+\tau_{photon}$ where *τ_RC_* is the *RC* constant of the electrodes, and $\tau_{photon}=\lambda_{0}Q/2\pi c_{0}$ is the photon lifetime in the micro-ring resonator. Here, *c*_0_, *λ*_0_, and *Q* are the vacuum light velocity, the ring resonance wavelength and the cavity quality factor, respectively.

According to static FEM simulations (see *C*/*L* in Table 1), we expect that that short cavity *τ_photon_* is generally much longer than the *RC* constant. As a result, the bandwidth is limited by the photon lifetime. In Figure 10 we show the applied voltage needed to satisfy the condition Δλ=1×δλ and the photon lifetime-induced bandwidth as a function of the cavity quality factor, for different SLOI platforms. In the simulations we have assumed an MRR with a ring radius of 50 μm [20] and the geometrical cross-section parameters as listed in Table 1. Although we have performed the simulations for all the considered platforms, not all curves are plotted in Figure 10 due to the overlapping among them. Here, the calculated bandwidth induced by the *RC* constant ranges from a minimum of 230 GHz for ${\left(\mathrm{ZnSe}\right)}_{2}/{\left({\mathrm{Ge}}_{2}\right)}_{7}$ to a maximum of 334 GHz for ${\left(\mathrm{AlN}\right)}_{2}/{\left(3C-\mathrm{SiC}\right)}_{7}$, confirming that the bandwidth of the electro-optic MRR modulator is limited by the photon lifetime. If fabrication of the MRR provides smooth waveguide walls, a *Q* of 20,000 should be achievable and in that case Figure 10 indicates modulation voltage in the 1.17–2.77 V range. If higher *Q* is attained, the required voltage is 0.5–1.0 V.

### 3.3. Broad Spectrum EO 2 × 2 MZI Crossbar Switches in the SLOI Platform

To move toward the large-scale integration of 2 × 2 switches in N × N switching matrices, we need to reduce the footprint of the device by selecting an active length of 50 μm or 200 μm. This would increase the switching voltage requirement on *V_s_* in a tradeoff condition. One way to constrain or reduce *V_s_* is to decrease the thickness of the upper oxide, that is, reducing *d*, which in turn increases IL via electrode loss. However, the smaller active length means that a higher loss per unit length is acceptable. Hence, we now chose 10 dB/cm for the switch investigation (see Figure 3).

Taking initially L = 200 μm, we then performed a set of simulations where the *V_π_L* minimization was obtained with a proper choice of *d* and *G*. Our results are presented in Table 2 for the selected best-performing platforms at the wavelength-of-operation shown.

The simulation work in Table 2 is summarized as *V_π_L* in the range 0.051–0.204 Vxcm, crossbar voltage 2.57–10.22 Volts, switching time less than 1 ps, switching energy 15 to 280 fJ/bit, and capacitance around 10 fF. At the telecom wavelength, the AlP/Si platform provides full switching at 9.8 Volts. Regarding large-scale integration for matrix switching, we note that the active length L could be reduced to 50 μm for the ZnSe/GaAs platform at the 764 nm wavelength, and in that case the crossbar applied voltage would be 10.4 Volts.

Table 1 and Table 2 modulation and switching results provided here indicate that our EO SLOI platforms are quite competitive with the leading state-of-the-art Pockels EO waveguided circuit platforms—ferroelectric [9] and poled organic polymer—in terms of Pockels coefficient value, switching voltage, footprint, speed, bandwidth, energy required, and *V_π_L*.

## 4. Multi-Function Superlattice Circuits

We have detailed the many Pockels possibilities for the SLOI PICs, and we wish to point out that a given Pockels PIC—any of the above seven SLOI PICs—also offers a strong second- or third-order nonlinear optical (NLO) response. In other words, a given PIC will perform NLO and EO functions at the same time, along with providing a complicated passive-waveguide circuit. Detector arrays are easily formed there as follows.

## 5. Integrated Photodetectors That Use One Superlattice Semiconductor

The SPSL is a “designed semiconductor”—a new, man-made semiconductor comprised of well-known semiconductors A and B, where the SPSL bandgap Eg(S) is the approximate average of Eg(A) and Eg(B). If we assume that Eg(B) is the smaller gap, then semiconductor B by itself would absorb all of the on-chip photons whose hν energy is Eg(S) < hν < Eg(B). We can put that absorption to use by growing a lattice-matched layer of B upon a strip waveguide in the A/B SPSL circuit, and by adding lateral P and N doping of the B rectangular upper-clad region. In that case, we have thereby formed a lateral PIN photodiode with evanescent-wave coupling to light traveling within the strip, since the refractive index of B is higher than that of the SPSL, causing automatic upward leakage of the guided mode into the PIN-B upper cladding volume. In other words, it will be easy to fabricate monolithic detector arrays in the SLOI PIC, and the detector material will be silicon for ZnS/Si, AlP/Si and GaP/Si platforms. It will be Ge for ZnSe/Ge and GaAs/Ge platforms, and it will be GaAs for the ZnSe/GaAs platform. Employing a similar procedure, monolithic avalanche photodiodes can be constructed on the SL waveguides in a lateral APD geometry.

## 6. Conclusions

In this paper, a physics-and-engineering procedure has been performed in order to investigate the Pockels coefficient in ${\left(\mathrm{GaP}\right)}_{N}/{\left({\mathrm{Si}}_{2}\right)}_{M}$, ${\left(\mathrm{AlP}\right)}_{N}/{\left({\mathrm{Si}}_{2}\right)}_{M}$, ${\left(\mathrm{ZnS}\right)}_{N}/{\left({\mathrm{Si}}_{2}\right)}_{M}$, ${\left(\mathrm{AlN}\right)}_{N}/{\left(3C-\mathrm{SiC}\right)}_{M}$, ${\left(\mathrm{GaAs}\right)}_{N}/{\left({\mathrm{Ge}}_{2}\right)}_{M}$, ${\left(\mathrm{ZnSe}\right)}_{N}/{\left(\mathrm{GaAs}\right)}_{M}$, and ${\left(\mathrm{ZnSe}\right)}_{N}/{\left({\mathrm{Ge}}_{2}\right)}_{M}$ short- and ultra-short-period superlattices. In this context, general physical aspects have been investigated by means of the empirical *sp*^3^*s** tight-binding method, by determining the features of the electronic-and-ionic structure and the influence of the monolayers number upon $\chi_{zzz}^{\left(2\right)}\left(\omega ,\omega ,0\right)$. We have explored large r_33_ coefficients in the proposed SL structures in the 500 nm–2000 nm wavelength range, demonstrating that efficient and “competitive” electro-optic modulation can be generated in both integrated broad-spectrum Mach-Zehnder interferometers and micro-ring resonators. We also investigated broadband 2 × 2 MZI crossbar spatial routing switches for N × N routing networks. The dominant Pockels coefficient in these SLOI platforms allows push-pull switching in all MZIs together with small footprint, low voltage, high speed, and energy efficiency.

Looking toward the future development of “on insulator” silicon-based photonic integrated circuits, which might include on-chip control electronics, the seven SLOI platforms proposed here feature a combination of desirable features: (1) a network of low-loss passive SL strip waveguides and components, (2) wide coverage of operation wavelengths, (3) strong nonlinear optical susceptibilities, (4) high-performance Pockels effect EO 1 × 1 modulators and 2 × 2 switches that offer a combination of excellent metrics, (5) simultaneous EO and NLO operation, and (6) monolithic lattice-matched on-chip SL waveguide-integrated photodetectors utilizing one of the two SL materials.

A wafer-bonding technique could be used to create the SLOI circuits just described. The theoretical work in this paper suggests that the undoped lattice-matched short-period superlattice platforms, in their seven-fold variety, could in the future offer cost-effective solutions to next-generation data center interconnections and to a variety of other important applications.

## Figures and Tables

**Figure 1 sensors-22-07866-f001:**
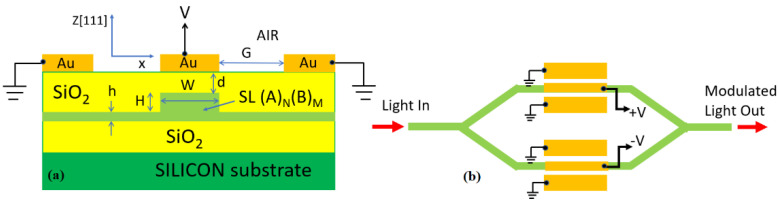
(**a**) Cross section view of a Pockels effect EO-waveguide modulator in the SLOI platform. (**b**) Top view of 1 × 1 EO amplitude-modulator based on the push-pull MZI architecture and Figure 1a waveguides.

**Figure 2 sensors-22-07866-f002:**
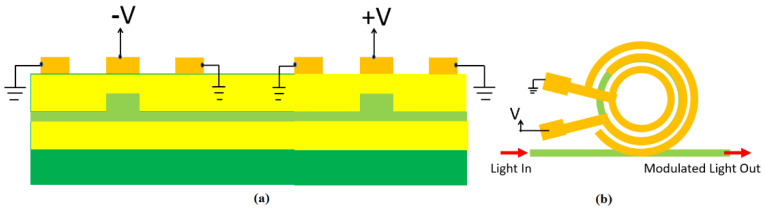
(**a**) Cross-section view of the active push-pull region in Figure 1b and in the 2 × 2 crossbar switch; (**b**) Top view of the resonant bus-coupled waveguided SLOI MRR EO amplitude-modulator showing the co-planar ring electrodes employed upon the ring waveguide and the ring rib regions.

**Figure 3 sensors-22-07866-f003:**
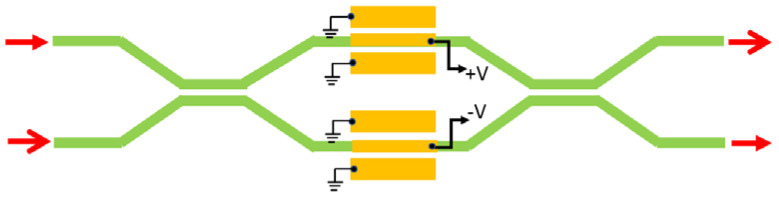
Top view of broad-spectrum 2 × 2 MZI crossbar switch in the SLOI platform. Electrodes on the localized push-pull region of Figure 2a are shown. The red arrows indicate the direction of light.

**Figure 4 sensors-22-07866-f004:**
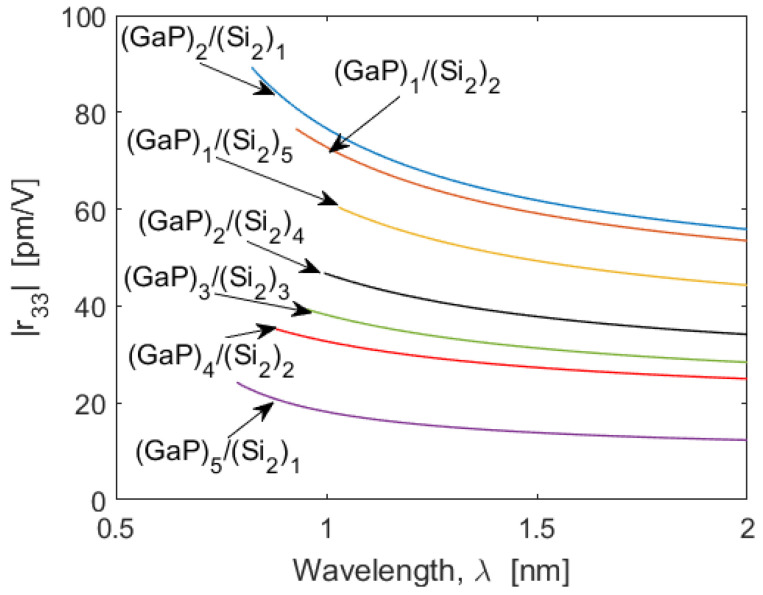
Pockels |r_33_| coefficient as a function of the wavelength, *λ*, for different ${\left(\mathrm{GaP}\right)}_{N}/{\left({\mathrm{Si}}_{2}\right)}_{M}$.

**Figure 5 sensors-22-07866-f005:**
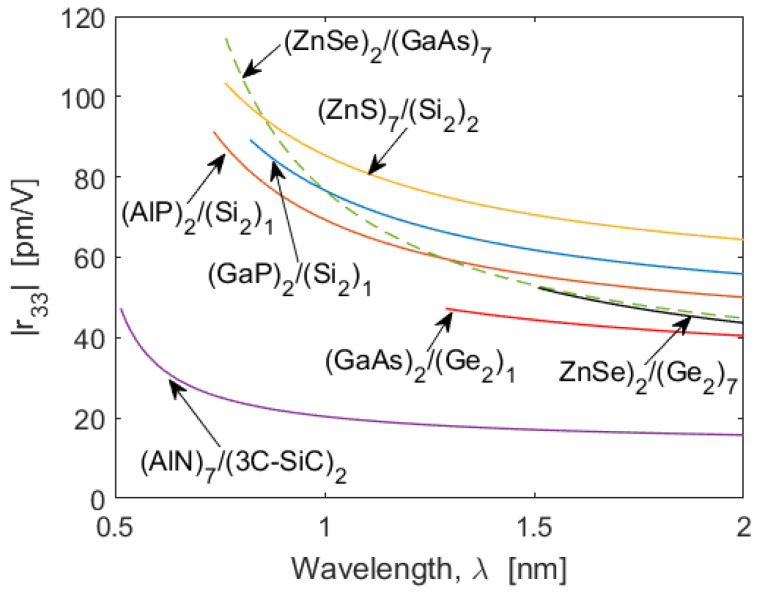
Pockels |r_33_| coefficient as a function of the wavelength, *λ*, for different optimized SL platforms. For each platform considered here, the left-side limit of each curve indicates its absorption edge. The numerical values of the wavelength edge are listed in Table 1.

**Figure 6 sensors-22-07866-f006:**
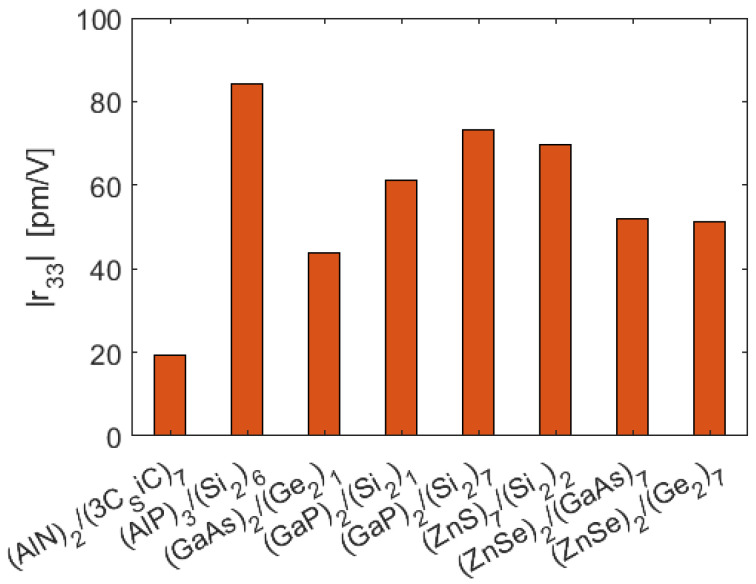
Pockels |r_33_| coefficient calculated at *λ* = 1550 nm telecoms for different optimized SL platforms. For the bulk materials constituting the SLs, the largest element of the Pockels tensor is: $\left|r_{41}\left(\mathrm{GaAs}\right)\right|$ = 1.6 pm/V, $\left|r_{41}\left(\mathrm{ZnS}\right)\right|\;$= 1.89 pm/V, $\left|r_{41}\left(\mathrm{GaP}\right)\right|$ = 1.1 pm/V, $\left|r_{41}\left(\mathrm{ZnSe}\right)\right|\;$= 2.0 pm/V, $\left|r_{33}\left(3C-\mathrm{SiC}\right)\right|\;$= 2.7 pm/V.

**Figure 7 sensors-22-07866-f007:**
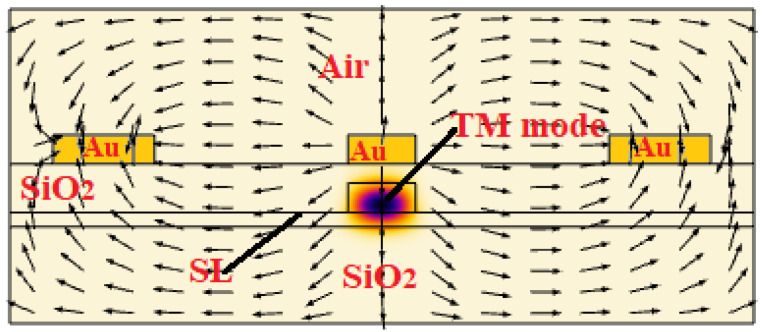
Simulated spatial distribution of the fundamental TM mode (color map) and applied static electric field (black arrows). In the simulation: *W* = 700 nm, *H* = 300 nm, *h* = 150 nm, *d* = 500 nm and SPSL: ${\left(\mathrm{GaP}\right)}_{2}/{\left({\mathrm{Si}}_{2}\right)}_{1}$.

**Figure 8 sensors-22-07866-f008:**
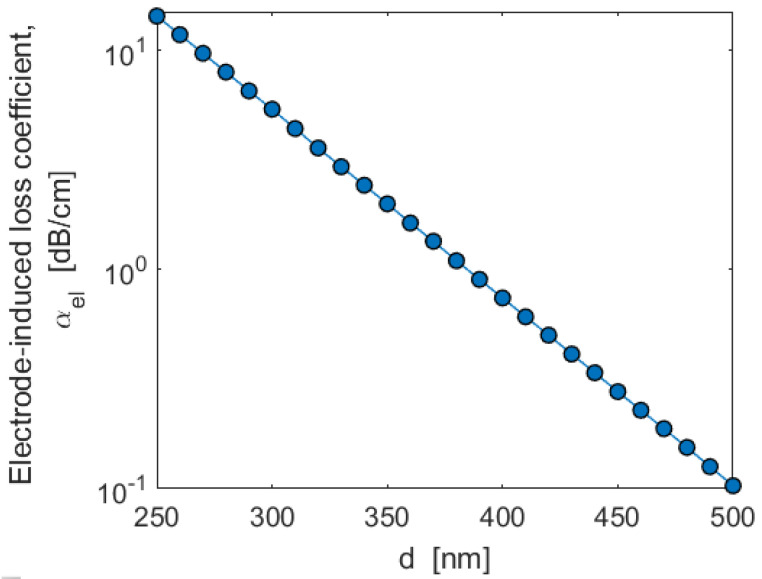
Simulated electrode-induced loss coefficient, *α_el_*, as a function of the parameter *d*. In the simulation: *W* = 700 nm, *H* = 300 nm, *h* = 150 nm, *G* = 2000 nm, *λ* = 822.5 nm, and SPSL: ${\left(\mathrm{GaP}\right)}_{2}/{\left({\mathrm{Si}}_{2}\right)}_{1}$.

**Figure 9 sensors-22-07866-f009:**
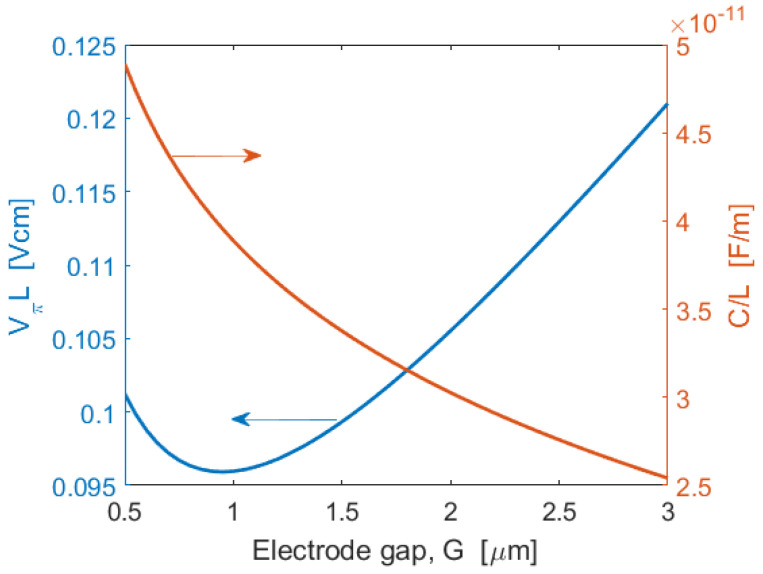
Simulated *V_π_L* and *C*/*L* as a function of the electrode gap *G*. In the simulation: *W* = 700 nm, *H* = 300 nm, *h* = 150 nm, *d* = 500 nm, *λ* = 822.5 nm, and SPSL: ${\left(\mathrm{GaP}\right)}_{2}/{\left({\mathrm{Si}}_{2}\right)}_{1}$.

**Figure 10 sensors-22-07866-f010:**
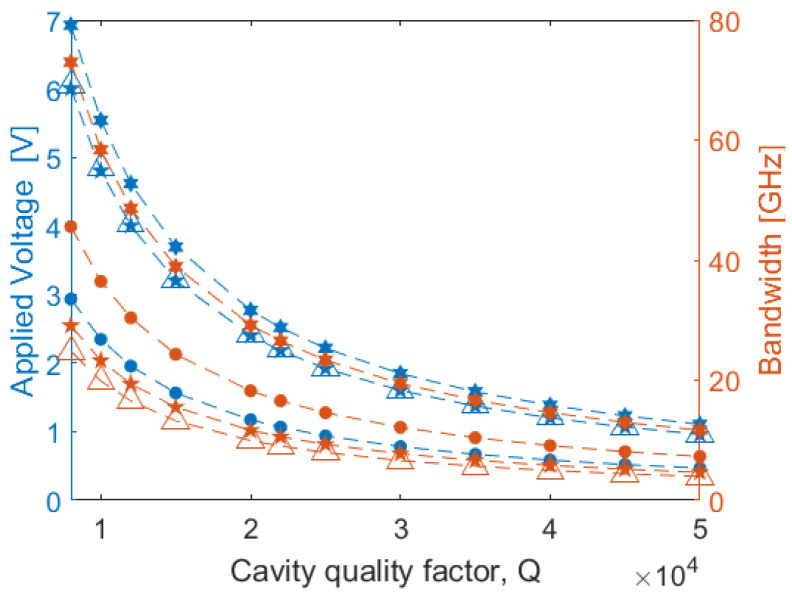
Applied MRR voltage, satisfying the conditions: Δλ=1×δλ and the photon-lifetime-induced bandwidth as a function of the cavity quality factor. Hexagram: ${\left(\mathrm{AlN}\right)}_{2}/{\left(3C-\mathrm{SiC}\right)}_{7}$; Pentagram: ${\left(\mathrm{GaAs}\right)}_{2}/{\left({\mathrm{Ge}}_{2}\right)}_{1}$; Triangle: ${\left(\mathrm{ZnSe}\right)}_{2}/{\left({\mathrm{Ge}}_{2}\right)}_{7}$; Circle: ${\left(\mathrm{GaP}\right)}_{2}/{\left({\mathrm{Si}}_{2}\right)}_{1}$.

**Table 1 sensors-22-07866-t001:** Figures of Merit of EO MZI push-pull 1 × 1 Modulators and 2 × 2 switches based on different SLOI Platforms. The active length is 1 mm for modulators and switches. The assumed insertion loss of the modulators and the switches is 1 dB per cm of active length. Figure 1a dimensions *d* and *G* are provided for optimized *V_π_L*.

SPSLs	*W* × *H* [nm] × [nm]	*λ* [nm]	Δ*E_v_* [eV]	|r_33_| [pm/V]	*V_π_L* [Vcm]	*V_s_* [V]	*C*/*L* [F/m]	*E_bit_* [fJ/bit]	*f*_3*dB*_ [GHz]
${\left(\mathrm{GaP}\right)}_{2}/{\left({\mathrm{Si}}_{2}\right)}_{1}$ *d* = 385 nm; *G* = 0.85 μm	700 × 300	822.5	0.24 [3]	89.27	0.08	0.8	4.105 × 10^−11^	6.57	77.54
${\left(\mathrm{AlP}\right)}_{2}/{\left({\mathrm{Si}}_{2}\right)}_{1}$ *d* = 355 nm; *G* = 0.80 μm	700 × 300	735.1	0.24 [3]	91.33	0.068	0.68	4.193 × 10^−11^	4.86	75.91
${\left(\mathrm{ZnS}\right)}_{7}/{\left({\mathrm{Si}}_{2}\right)}_{2}$ *d* = 410 nm; *G* = 1.0 μm	700 × 400	762.2	0.70 [22]	103.5	0.10	1.00	3.92 × 10^−11^	9.9	81.14
${\left(\mathrm{ZnSe}\right)}_{2}/{\left(\mathrm{GaAs}\right)}_{7}$ *d* = 360 nm; *G* = 0.80 μm	700 × 300	763.8	0.72 [23]	114.7	0.062	0.62	4.214 × 10^−11^	4.08	75.53
${\left(\mathrm{ZnSe}\right)}_{2}/{\left({\mathrm{Ge}}_{2}\right)}_{7}$ *d* = 392 nm; *G* = 1.0 μm	900 × 600	1510	1.12 [24]	52.4	0.275	2.75	4.40 × 10^−11^	83.38	72.35
${\left(\mathrm{GaAs}\right)}_{2}/{\left({\mathrm{Ge}}_{2}\right)}_{1}$ *d* = 395 nm; *G* = 0.95 μm	900 × 550	1290	0.31 [25]	47.24	0.244	2.44	4.38 × 10^−11^	65.31	72.68
${\left(\mathrm{AlN}\right)}_{7}/{\left(3C-\mathrm{SiC}\right)}_{2}$ *d* = 391 nm; *G* = 1.1 μm	400 × 220	513.8	0.65 [26]	47.29	0.12	1.2	3.035 × 10^−11^	11.03	104.9

**Table 2 sensors-22-07866-t002:** Figures of Merit of EO MZI broadband push-pull 2 × 2 Crossbar Switch based on different SLOI Platforms. The active length is 200 μm for these Figure 3 switches. The assumed insertion loss of the switches is 10 dB per cm of active length. Figure 1a dimensions *d* and *G* are provided for optimized *V_π_L*.

SPSLs	*W* × *H* [nm] × [nm]	*λ* [nm]	*V_π_L* [V × cm]	*V_s_* [V]	*C* [fF]	*E_bit_* [fJ/bit]	Switch Time [ps]
${\left(\mathrm{GaP}\right)}_{2}/{\left({\mathrm{Si}}_{2}\right)}_{1}$ *d* = 268 nm; *G* = 0.65 μm	700 × 300	822.5	0.065	3.27	8.98	23.98	0.449
${\left(\mathrm{GaP}\right)}_{2}/{\left({\mathrm{Si}}_{2}\right)}_{7}$ *d* = 295 nm; *G* = 0.7 μm	1200 × 700	1550	0.204	10.22	10.75	280.7	0.537
${\left(\mathrm{AlP}\right)}_{2}/{\left({\mathrm{Si}}_{2}\right)}_{1}$ *d* = 251 nm; *G* = 0.7 μm	700 × 300	735.1	0.057	2.83	8.78	17.64	0.439
${\left(\mathrm{AlP}\right)}_{3}/{\left({\mathrm{Si}}_{2}\right)}_{6}$ *d* = 315 nm; *G* = 0.75 μm	1200 × 700	1550	0.196	9.8	10.52	252.9	0.525
${\left(\mathrm{ZnS}\right)}_{7}/{\left({\mathrm{Si}}_{2}\right)}_{2}$ *d* = 287 nm; *G* = 0.80 μm	700 × 400	762.2	0.082	4.12	8.47	36.05	0.423
${\left(\mathrm{ZnSe}\right)}_{2}/{\left(\mathrm{GaAs}\right)}_{7}$ *d* = 253 nm; *G* = 0.65 μm	700 × 300	763.8	0.051	2.57	9.05	15.0	0.453

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
