# Peer review of "High-Performance Pockels Effect Modulation and Switching in Silicon-Based GaP/Si, AlP/Si, ZnS/Si, AlN/3C-SiC, GaAs/Ge, ZnSe/GaAs, and ZnSe/Ge Superlattice-On-Insulator Integrated Circuits"

_sensors, 2022, doi:10.3390/s22207866_

Round 1

Reviewer 1 Report

The author well presented non-linear properties of various super lattices(SLs).

It could be interesting for possibility of improved non-linear based photonic devices.

However, such SL platforms (GaP/Si, AlP/Si, ZnS/Si, AlN/3C-SiC, GaAs/Ge, ZnSe/GaAs, and ZnSe/Ge) seem to be impossible to growth epitaxially and to fabricate with wafer bonding, at present.

The author should add the reason and motivation to choose such kinds of materials for SL.

Author Response

Here we attach the reply file 

Reviewer 2 Report

The authors give a thorough theoretical study of electro-optic properties of novel hypothetical superlattice materials consisting of alternating III-V/II-VI/IV monolayers. Device-level simulations are provided and figures of merit are calculated. The work is of high scientific interest since there are numerous shortcomings with current EO materials used in photonic ICs (Si modulators have high insertion loss, LiNbO3 has low r33 and is not CMOS-compatible, BaTiO3 epitaxy and integration is challenging, etc). Proposed superlattice materials offer low insertion loss (assuming high quality epitaxy), low voltage per pi phase shift, low power consumption, and high speed. In my personal opinion, such superlattice materials could enhance PIC performance when heterogeneously integrated on mature Si and SiN platforms. I recommend the article for publication after authors address couple minor comments:

1) Line 102. Since authors mentioned a perspective to grow superlattice materials on 300 mm wafers, I assume PIC fabrication should take place in CMOS foundries. Therefore, would be nice to see device simulations with CMOS-compatible electrode metals, like W, Cu or Al. If metal choice (Au vs CMOS metal) has a minor/negligible effect on device FoMs, would be good to write a line mentioning that CMOS-compatible metals will yield similar modulator performance.

2) Line 128. Superlattice materials are benchmarked against BaTiO3 and LiNbO3, would be good to add a reference with recent progress on BaTiO3 modulators.

3) In Figure 9 minimal VpL is found at the waveguide-electrode distance of ~1000 nm. What would be the explanation that closer distance results in higher VpL? At the first glance, electric field in waveguide should be stronger when electrodes are closer.

Author Response

Here we attach the reply file

Reviewer 3 Report

Seven new superlattice-on-insulator structures are proposed by the authors for potential applications in waveguided electro-optical (EO) modulators. Sufficient backgrounds and calculation details are provided by the authors and the work is interesting. However, several issues need to be addressed before it can be accepted for publication in Sensors.

1. The presented work assumes a "nearly perfect" structure by omitting fabrication errors. Although it is acceptable to ignore material imperfections and etching errors in this theoretical work, the presence of strain is unavoidable in the superlattice structure proposed in this work. The authors should discuss how they expect the strain affects the Pockels coefficient.

2. In Figure 5, the authors plot the Pockels coefficient as a function of the wavelength for the seven proposed superlattice structures. The absorption edge of the constituting material of each superlattice should be labeled or indicated in the figure as a reference since mostly the wavelength above this edge will be of interest for application purposes.

3. The authors claimed the superlattice shows a larger effect than the bulk material although it is not clear how much enhancement was achieved. The Pockels coefficients of the bulk materials should be plotted together with the superlattice structure in Figure 6.

4. On page 2, line 60, the authors mentioned that smooth nano-layering was achieved before. Some references should be cited, Wu et al., Appl. Phys. Lett. 116, 013101 (2020), https://doi.org/10.1063/1.5124828; Ahn et al., Science, 303, 488 (2004), DOI: 10.1126/science.1092508; Jackson et at., J. Crystal Growth, 270, 301 (2004), DOI:10.1016/j.jcrysgro.2004.06.033.

5. There are typos and missing words in the context, the authors should correct them.

Author Response

Here we attach the reply file
